

# Levels of biomarkers associated with subconcussive head hits in mixed martial arts fighters

Nelson Marinho de Lima Filho[1], Sabrina Gabrielle Gomes Fernandes[1], Valeria Costa[2], Daline Araujo[3], Clecio Godeiro Jr[4], Gerlane Guerra[5], Ricardo Oliveira Guerra[1] and Karyna Figueiredo Ribeiro[1]

[1] Department of Physiotherapy, Federal University of Rio Grande do Norte, Natal, Brazil
[2] Health Science Center, Postgraduate Program in Drug Development and Technological Innovation, Federal University of Rio Grande do Norte, Natal, Brazil
[3] Health Sciences College of Trairi, Federal University of Rio Grande do Norte, Santa Cruz, Brazil
[4] Department of Integrated Medicine, Federal University of Rio Grande do Norte, Natal, Brazil
[5] Department of Biophysics and Pharmacology, Federal University of Rio Grande do Norte, Natal, Brazil

Corresponding author
Sabrina Gabrielle Gomes Fernandes,
sabrinaggf@hotmail.com

## ABSTRACT

**Background**. Concussion and the damage resulting from this event related to brain function have been widely studied; however, little is known about subconcussive impacts, especially in Mixed Martial Arts (MMA) fighters, which is a combat and full contact sport in which most blows are aimed at the head.

**Objective**. This study aims to evaluate the biomarker levels associated with subconcussive hits to the head in MMA fighters.

**Methods**. This is an exploratory study in which 30 male subjects (10 MMA fighters, 10 healthy individuals who practice muscle training, and 10 healthy sedentary individuals) aged between 18 and 32 years ($25.4 \pm 3.8$) were evaluated. These individuals underwent blood collection to assess their Ubiquitin C-terminal hydrolase (UCH-L1), Glial Fibrillary Acidic Protein (GFAP) and Brain Derived Neurotrophic Factor (BDNF) levels before, immediately after and 72 hours after the sparring session (for the fighters) and were compared between groups.

**Results**. Significant differences were found at baseline between active and healthy fighters in BDNF levels ($p = 0.03$). A significant reduction of BDNF levels were also observed between the post-immediate and 72h after the sparring session ($p = 0.03$). No differences were observed in the number or severity of symptoms reported by the fighters.

**Conclusion**. Despite the exploratory approach, the findings of this study may help to understand the influence of repeated subconcussive hits to the head in MMA fighters, as well as to propose preventive interventions which can minimize the effects of the impact of hits, preserving fighters' neuronal integrity and function.

## INTRODUCTION

Sports-related concussion (SRC) is a traumatic brain injury caused by a direct blow to the head, neck or body resulting in an impulsive force being transmitted to the brain that occurs in sports and exercise-related activities (*Patricios et al., 2023*), this initiates a neurotransmitter and metabolic cascade, with possible axonal injury, blood flow change and inflammation affecting the brain. This event causes rapid acceleration and deceleration; however, the sub-concussive impacts do not result in a known or diagnosed concussion on clinical grounds (*Hunter, Branch & Lipton, 2019*).

According to prior studies, the rate of concussions varies according to age and the type of sport practiced (*Gardner & Yaffe, 2015*; *Pfister et al., 2016*) for example, a meta-analysis of incidence rates of concussion in professional rugby union players ($n = 16$ studies) reported an incidence rate of 1.19 concussions per 1,000 player-match hours (range, 0.33–7.8) (*Gardner et al., 2014*). In Mixed Martial Arts (MMA), that the rate of head injuries is higher, many studies have attempted to determine the prevalence of concussions in MMA, with variable ranges of 8.3% to 62.3 (*Hada et al., 2022*; *Hamdan et al., 2022*), with jiu-jitsu Brazilian and wrestling accounting for 25.2% and 19.5% of these events, respectively (*Karpman et al., 2016*; *Miarka et al., 2022*).

With emerging evidence of both short- and sometimes long-term consequences following concussion, concern for the health of combat sport athletes has become a priority (*Brown et al., 2021*). Clinically, SRC commonly results in neurological impairment for a short period of time and is usually spontaneously resolved following a sequential course (*McCrory et al., 2017*). Symptoms such as headaches, dizziness, and neck pain, are also frequently reported by individuals with cervical spine and vestibular system dysfunction following head and neck trauma.

Long-term consequences of SRC can occur, ranging from balance problems, attention deficit, sleep disturbances, and frequent irritability, to suicide attempt(s) (*Gardner & Yaffe, 2015*). There is evidence that severe or moderate traumatic brain injury may represent an important risk factor for degenerative diseases such as Alzheimer's, Parkinson's (*Gardner & Yaffe, 2015*), and frontotemporal dementia (*Kalkonde et al., 2012*; *Rosso et al., 2003*; *Morissette et al., 2020*), as suggested by previous studies showing to repetitive head impacts, such as that experienced by some professional athletes, as potentially associated with the development of the specific neuropathology (*Katz et al., 2021*).

Among brain traumas, subconcussion can be highlighted. Defined as "a cranial impact which does not result in a known or diagnosed concussion in the clinical field" (*Bailes et al., 2013*), the subconcussion does not cause symptoms, and is less severe than a concussion, so it usually does not result in a clinical diagnosis (*Díaz-Rodríguez & AP, 2019*) Despite advances in instrumentation in clinical practice, SRC remains difficult to diagnose. The challenges for differential diagnosis are the absence of biomarkers and paucity of clinical tests specific to concussion. As a result, tools developed to assess other conditions have been borrowed and adapted to evaluate many of the effects of concussion (*Mucha & Trbovich, 2019*).

Wording it is currently considered that concussions should also be assessed by physical examinations which include assessments of mental health, like cognitive impairment and mood disturbances (anxiety and depression) (*Hutchison et al., 2018*), balance, reflex testing, and coordination (*Stillman et al., 2017*). It is also important to consider analysis of relatively inexpensive blood biomarker levels to measure analytes linked to pathophysiological processes such as inflammation and neurodegeneration as a diagnostic tool (*Di Battista, Rhind & Baker, 2013*; *Sahu et al., 2017*).

There is growing interest in studies on the serum evaluation of concussion-related biomarkers, as it plays a critical role in characterizing both severity and duration of injury (*Agoston, Shutes-David & Peskind, 2017*), in addition to supposedly capturing various aspects of the neurometabolic cascade of concussion, as, according to recent findings, some of the proposed biomarkers for SRC are sensitive to exercise and/or exposure to small impacts on the head during participation in sports (*Meier et al., 2017a*; *Joseph et al., 2019*; *Di Battista et al., 2018*)

Serum brain injury biomarkers are usually proteins released by neuronal or glial cells when subjected to pathological changes. Two of the most promising brain injury marker proteins are glial fibrillary acidic protein (GFAP) and ubiquitin C-terminal hydrolase (UCH-L1). GFAP is a marker of astroglial injury and is found in the astroglial skeleton of the white and gray matter of the brain, and whose functions include cellular communication, mitosis and maintenance of the integrity of the blood–brain barrier (BBB) (*Hol & Pekny, 2015*), while UCH-L1 is a marker of neuronal injury23, which presents high levels in patients with traumatic brain injury of all categories, and its expression may be associated with the severity of the injury (*Meier et al., 2017b*).

Another important biomarker related to SRC is the Brain Derived Neurotrophic Factor (BDNF), a neurotrophic protein which is autocrine secreted and promotes neuron development, maintenance, survival, regeneration, and differentiation (*Cohen-Cory et al., 2010*), and is also important for synaptic plasticity and memory processing (*Alonso et al., 2002*; *Bekinschtein et al., 2008*). According to *Kaplan, Vasterling & Vedak (2010)*, BDNF is an appealing candidate biomarker for detecting TBI for numerous reasons, mainly because a very strong association between BDNF and trauma brain injury, yielding excellent discriminative ability of 0.94–0.95 (as measured by the c-statistic) (*Korley et al., 2016b*).

Numerous publications from the ongoing Professional Fighters Brain Health Study and other previous studies have demonstrated an association of increased changes of the brain on imaging and increased cognitive impairment with higher levels of fight exposure (frequency of competing and duration of career) in combat sports athletes (*Mishra et al., 2017*; *Shin et al., 2014*; *Ravdin et al., 2003*). Retrospective studies conducted by *Jordan et al. (1996)* and *Kaste et al. (1982)* show neuropsychological deficits in memory tests, information processing speed, complex attention, and executive functioning in fighters. Associated with this, repetitive head traumas have been considered a risk factor for neurodegenerative disorders such as depression and chronic traumatic encephalopathy (*Bazarian et al., 2009*).

*Bernick et al. (2021)*, who evaluated the number of concussions that occur during a combat sport competition, it was observed that of the 60 fights analyzed, 47 involved blows

to the head, of which 43% occurred in MMA. Due to increasing number of amateurs and professionals MMA practitioners, it is necessary to understand the influence of multiple subconcussive hits to the head on MMA fighters, as well as their relationship with the biomarker levels related to concussion and symptomatology. Most of the strikes as well as impacts from landing/takedowns in MMA fights are directed to the head (*Buse, 2006*), and cumulative exposures to repeated blows to the head may cause damage.

Although concussions and their brain function-related damage has been widely studied in many sports, little is known about the effects of subconcussive hits to the head (*Gysland et al., 2012*), but it is already known that even sub-concussive hits are sufficient to lead to the development of degenerative brain disease (*Stein, Alvarez & McKee, 2015*; *Baugh et al., 2012*).

Thus, due to the scarcity of studies in the literature, in addition to the need to use effective diagnostic tools to elucidate the clinical conditions resulting from combat sports injuries, we hypothesize that successive blows to the head during MMA practice can lead to a significant increase on levels of biomarkers associated with subconcussion. Therefore, the aim of this study to evaluate the physiological response of biomarkers with subconcussive hits to the head in MMA fighters.

## METHODS

### Participants

This exploratory study enrolled 40 participants aged between 18 and 32 years (25.4 ± 3.8) carried out between May and August 2019. The participants were divided into three groups: (1) MMA fighters; (2) Active control group: active individuals according to the International Physical Activity Questionnaire –IPAQ, an instrument for measuring the physical activity level (*IPAQ Research Committee, 2005*), and more precisely muscle training practitioners; (3) Sedentary control group: healthy sedentary individuals according to the Brazilian version of the Sedentary Behavior Questionnaire (*Mielke GI, Owen & PC, 2014*). Thus, a total of 16 MMA fighters from three training centers of a city in Northeast Brazil compared with 14 resistance training practitioners, and 11 healthy sedentary men recruited through social media (Fig. 1).

### Ethics

Study procedures were approved in December, 2017 by the Research Ethics Committee of the Federal University of Rio Grande do Norte (protocol reference #3.246.228), and all participants provided written informed consent prior to the study.

### Procedure

A member of the research team (graduation and post-graduation students) provided information about the study and requested consent to drain blood samples and use the Sports Concussion Assessment Tool 5th edition (SCAT5) results for research purposes.

For the group of fighters, the collections were carried out in a controlled environment, during the sparing session, which is fight training involving two individuals under the supervision of an experienced trainer or technician. For the present data collection, only

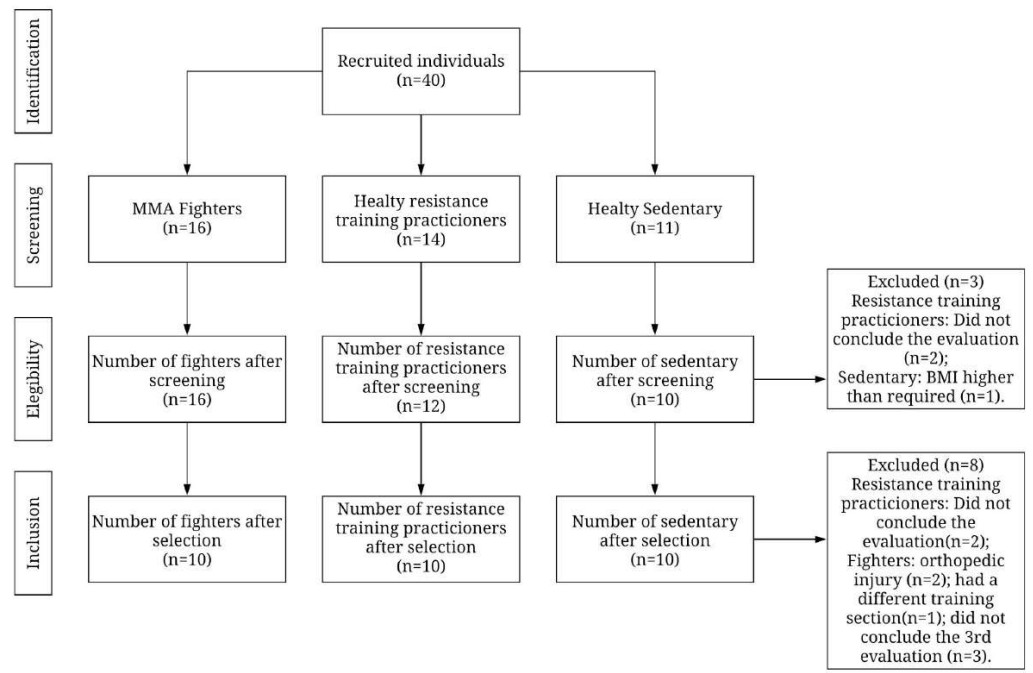

**Figure 1** **Flowchart of subject recruitment.**

one session was carried out, lasting 3-4 rounds of 5 min each. As for the equipment used during the fight, the participants only wore gloves specific to MMA, with protection for the wrist and finger joints.

The individuals of the control group (Healthy Active Group) were instructed not to exercise on the data collection day for all the following tests, and the fighters were instructed not to participate in any contact training during the study period. The group of healthy sedentary individuals was instructed to maintain their normal routine of daily activities. The assessments (SCAT5 and blood collection) were carried out at three moments: immediately before the fight, up to 30 min after and 72 h after the fight for fighters and at a single moment for active and sedentary individuals.

## Covariates

Sociodemographic and anthropometric data of the individuals (age, education level height, weight, and body index mass (BMI), lifestyle habits (smoking history and alcohol intake), were evaluated by self-report, and physical activity frequency were collected in the baseline assessment.

## Concussion symptoms evaluation

Sports Concussion Assessment Tool 5 (SCAT5): the SCAT5 is a standardized questionnaire to evaluate athletes with concussion. It can be used to evaluate children from 13 years old (*Patricios et al., 2023*). The questionnaire consists of 6 evaluation steps; however, this study only used the first part of the tool regarding the evaluation of the number and severity of the symptoms, according to the studies of *Meier et al. (2017a)* and *Di Battista*

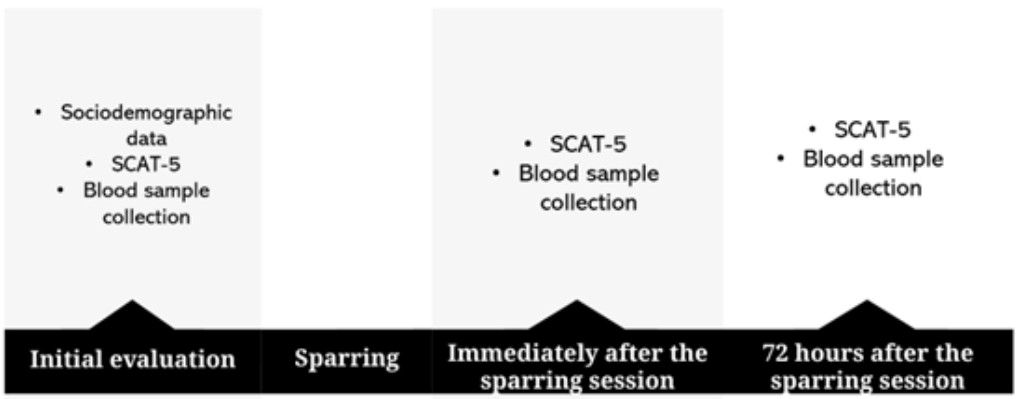

**Figure 2**  **Summary of assessments that occurred at each time of the study.**

*et al. (2018)*. The translation of the symptoms scale performed and used by the Brazilian Football Confederation was used for this purpose. The SCAT5 was applied three times (see Fig. 2).

The symptoms score is composed of 22-item symptoms using a seven-point Likert scale rating. The severity of the symptoms is obtained by the sum of the rated symptoms score (*Guskiewicz et al., 2013*). This scale has shown to be reliable for assessing symptom presence and severity (*Luoto et al., 2019*).

## Blood sample collection

Eight milliliters of venous blood were drawn from the participants by a certified nurse using EDTA K 2+ tubes. The blood collection was done at three time points. After a short cooling down period, the samples were stored in a cooler with ice at a temperature of around 17 °C. The samples were centrifuged shortly after in a centrifugal machine (CentriBio 80-2B) at about 2,500 revolutions per minute for 10 min in order to separate the blood plasma. The plasma was stored in microtubes in aliquots of 500 µl and maintained at a temperature of −80 °C until the sample analysis. Blood samples were all collected at the same time of the day, in the afternoon, and none of the participants were fasting. At the time of the first blood collection, for all fighters was requested to avoid any head impacts or head trauma for 5 days prior, following the researchers' recommendations.

## Biomarker analysis

Plasma concentration of three blood biomarkers (UCH-L1, GFAP and BDNF) were determined using the Enzyme-linked Immunosorbent Assay (ELISA) from Elabscience (Houston, TX). First, duplicate samples of 100 µl of each subject were used following the manufacturer's instructions. The reading was performed at 450 normetanephrine (nm) in an ELISA reader (Mindray MR-96A) and the results were interpolated into a standard curve for each marker.
### Statistical analysis

Quantitative variables are presented for central tendency (mean) and dispersion measures (standard deviation), while the categorical variables were expressed for absolute and relative frequencies. The Shapiro–Wilk test was utilized to check the data normality. The difference between groups related to the categorical variables was assessed by the Qui-square test (for descriptive analysis). A one-way ANOVA with Tukey's post hoc was performed to compare the biomarker levels between fighter group and healthy active and sedentary groups in the initial evaluation. After that, ANOVA for repeated measures was performed to analysis of the difference between the pre-sparring moment, immediately after and 72 h after the sparring session, the number and severity of symptoms and levels of concussion-related biomarkers (only for the fighter group). Effect size (Cohen's d) was calculated for biomarkers values between groups (in baseline). Values between 0.20 and 0.49 were considered small effect size, 0.50 to 0.79 represented moderate size, and over 0.79 was considered as a large size (*Cohen, 1988*).

All data were analyzed using the GraphPad Prism version 8.2 software program (GraphPad Inc., La Jolla, CA, USA). A statistical significance of 5% was established for all analyzes.

## RESULTS

### Sample characterization

A total of 30 male individuals were included in this study after the selection and meeting the inclusion criteria (10 fighters, 10 resistance training practitioners, and 10 healthy sedentary men). Participant characteristics, socio-demographic data, and concussion symptoms and severity are listed in Table 1. There was no difference between the groups related to weight, height, and BMI. The fighters had a mean of $6.4 \pm 3.2$ years' experience of professional practice with a frequency of $13.4 \pm 0.8$ h training sessions per week ($p < 0.0001$). In the variables related to concussion, evaluated through the SCAT5 at the three moments, the number of symptoms before the fight was higher in the group of fighters (SCAT5 - Number of Symptoms = 6) when compared to both control groups (SCAT5 - Number of Symptoms = 0.9 and 4.1 for Active control and sedentary group, respectively) ($p = 0.01$). The fighter's group had a greater severity of symptoms before the fight (SCAT5 = 11.3) compared to the active healthy group (SCAT5 = 1.3) and sedentary healthy group (SCAT5 = 7.2), with a statistically significant difference ($p < 0.01$).

When comparing the number and severity of symptoms, as measured by the SCAT-5, only in the group of athletes across the three assessment times, no significant differences were found ($p$-value: 0.73 for number of symptoms and p: 0.86 for severity of symptoms).

### Biomarker analyzes

In assessing BDNF, it was possible to identify differences between the fighters and active control in baseline ($p = 0.03$) (Fig. 3A). There were no significant differences between groups for UCH-L1 or GFAP markers during the first assessment (Figs. 4A and 5A). It was further possible to identify a significant difference between the immediate and 72 h post sparring moments when evaluating BDNF ($p = 0.03$) (Fig. 3B). However, for UCH-L1

**Table 1 Sample characteristics ($N = 30$).**

| | Fighters | | | Active control | Sedentary control | |
| --- | --- | --- | --- | --- | --- | --- |
| | $n = 10$ | | | $n = 10$ | $n = 10$ | p-value |
| | | Mean (SD) or % | | | | |
| Age | 24.20 (4.5) | | | 27 (2.6) | 25.10 (3.8) | 0.25[a] |
| Weight (Kg) | 76.30 (11.2) | | | 72 (8.5) | 73.60 (10.4) | 0.64[a] |
| Height (cm) | 173.80 (7.6) | | | 174.30 (5.3) | 171.50 (5.0) | 0.56[a] |
| BMI (Kg/m$^2$) | 24.85 (1.8) | | | 23.66 (2.2) | 24.94 (2.8) | 0.41[a] |
| Education Level | | | | | | 0.03[b*] |
| High School | 70% | | | 10% | 20% | |
| Higher Education | 30% | | | 50% | 50% | |
| Post-graduation | - | | | 40% | 30% | |
| Smoking | | | | | | 0.58[b] |
| Yes | 10% | | | 10% | – | |
| No | 90% | | | 90% | 100% | |
| Drinking | | | | | | 0.47[b] |
| Yes | 70% | | | 90% | 70% | |
| No | 30% | | | 10% | 30% | |
| Physical Activity Frequency (activities per week) | 13.4 (0.8) | | | 4.6 (0.4) | – | <0.0001[a*] |
| SCAT-5 | Time 01 | Time 02 | Time 03 | | | |
| Number of Symptoms[f1] | 6.0 (4.8) | 6.5 (4.4) | 3.9 (3.0) | 0.9 (1.1) | 4.1 (3.9) | 0.01[c#] |
| p-value[e] | 0.73 | | | – | – | – |
| Severity of Symptoms[f2] | 11.3 (8.4) | 10.1 (6.9) | 5.5 (5.3) | 1.3 (1.9) | 7.2 (5.7) | <0.01[#'c] |
| p-value[e] | 0.86 | | | – | – | – |

Notes.

(Kg), Kilograms; (Kg/m$^2$), Kilograms per square meter; BMI, Body Index Mass.

[a] p-value for ANOVA one way test

[b] p-value for Qui-square test

[c] p-value for ANOVA for repeated measures.

[*] Education level: There is a difference between the fighting group and the other groups.

[#] Frequency of physical activity (practices a week): There is a difference between active and healthy fighting groups.

[#'] Symptom assessment: There is a difference between the fighter group and the other groups before (time 01), immediately after (time 02), and 72 h after the sparring session (time 03).

[e] p-value for ANOVA for repeated measures.

[f] f1. $df = 2$; $F = 6.70$; f2. Df=2; $F = 6.70$.

levels there was a significant difference between pre-sparring and immediately post-sparring ($p = 0.04$) (Fig. 4B). There were no significant differences between the different assessment times for the GFAP levels (Fig. 5). Differences in BDNF levels were observed between the fighters immediately after the sparring session and the active control group ($p = 0.01$) (Fig. 3C), however, we did not find a difference between 72 h post-sparring and the control groups (Fig. 3D). No differences were found in UCH-L1 or GFAP levels (Figs. 4C and 4D; 5C and 5D). The levels of the other proteins revealed no differences between groups. The values of each biomarker, according to the groups, are available in Table S1. It is possible to observe in S1 the results of effect size (ES) between biomarker measurements. For BDNF the effect size was large, while ES for UCHL-1 was small. It was not possible to calculate the ES for GFAP.

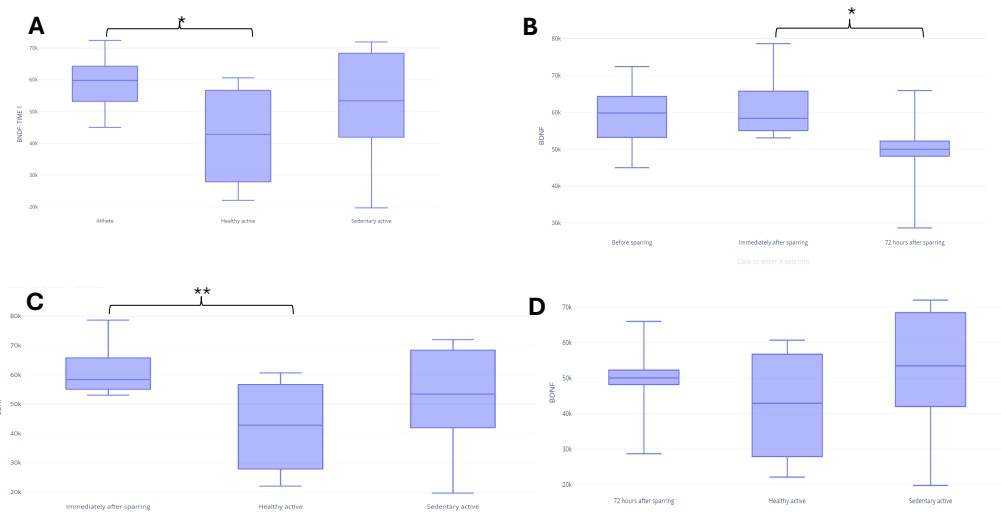

**Figure 3** **Differences in BDNF between groups and assessment time points.** Note: (A) Difference between the evaluated groups*; (B) difference between the three assessment time points in the athlete group. (C) difference in BDNF identified immediately after sparring and control groups**; (D) difference in BDNF identified 72 h after sparring and control groups. Unit: picograms per milliliter (pg/mL). *($p < 0,05$); ** ($p < 0.001$).

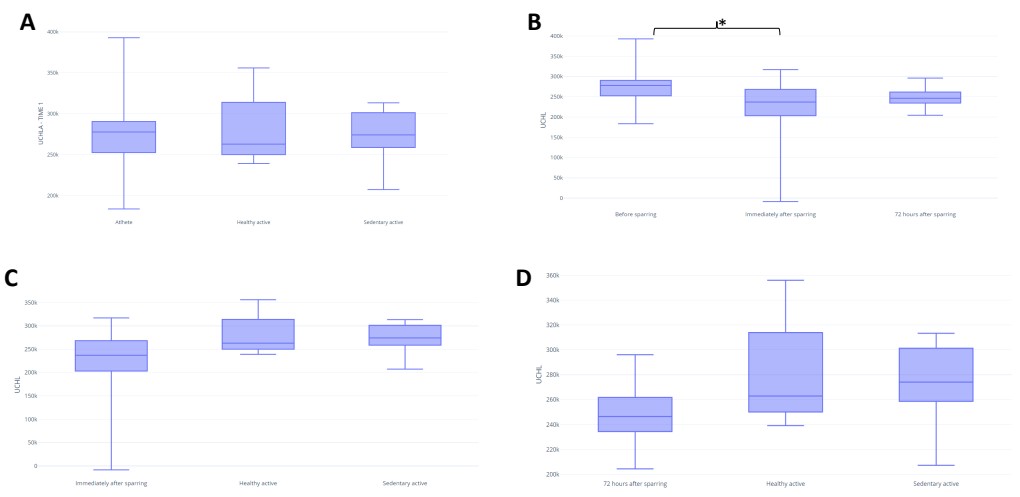

**Figure 4** **Differences in UCHL between groups and assessment time points.** Note: (A) Difference between the evaluated groups; (B) difference between the three assessment time points in the athlete group*. (C) difference in UCHL identified immediately after sparring and control groups; (D) difference in UCHL identified 72 h after sparring and control groups. Unit: picograms per milliliter (pg/mL). *($p < 0,05$).

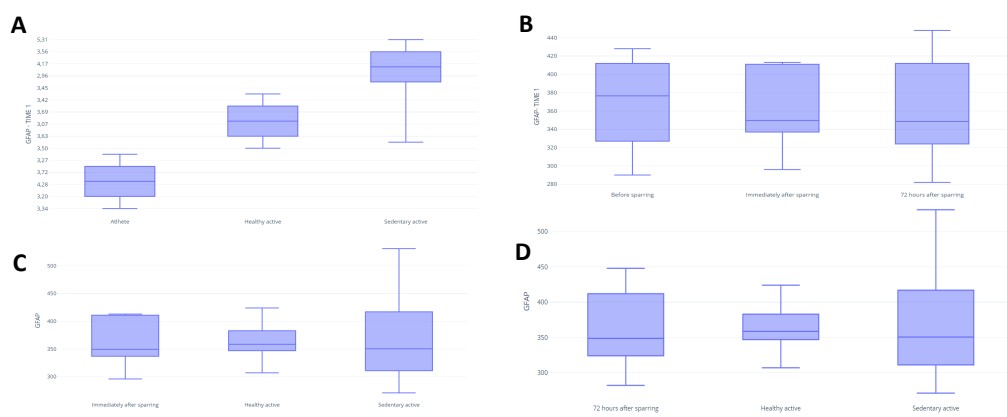

**Figure 5** **Differences in GFAP between groups and assessment time points.** Note: (A) Difference between the evaluated groups; (B) difference between the three assessment time points in the athlete group. (C) difference in GFAP identified immediately after sparring and control groups; (D) difference in GFAP identified 72 h after sparring and control groups. Unit: picograms per milliliter (pg/mL).

## DISCUSSION

Our study pioneered the effects of repeated subconcussive hits to the head on concussion-associated protein levels in MMA fighters. Our results observed that the group of fighters presented a greater number and severity of symptoms in the SCAT5 when compared to the other groups, regardless of the evaluation time. Regarding biomarker analysis, BDNF and UCHL-1 were significant among the groups evaluated, being higher in the group of fighters. For better understanding, the discussion of this study will be presented in topics considering the evaluated biomarkers.

### BDNF

The use of BDNF as a marker of the effects of physical activity and exercise is well documented (*Huang et al., 2014*). In addition, BDNF is clinically important due to its role in neuronal integrity because it expresses functional status (memory processing) and regenerative power (synaptic plasticity) (*Cohen-Cory et al., 2010*). An assessment of serum levels can be an important diagnostic tool and help to clinically manage alleged subconcussive scenarios of individuals exposed to trauma.

High BDNF levels were observed in the fighting group when compared to the active group in the initial assessment in our study (588.5 (79.0) *vs* 423.9 (144.3), respectively), like Oztasyonar (*Oztasyonar, 2017*) who found significantly higher post-training BDNF levels in the taekwondo fighters, boxers, and athletes groups compared to pre-training level. These findings may be supported by the type and level of physical activity intensity practiced by the groups (*Oztasyonar, 2017*). The active group practiced strength exercises; on the other hand, the group of fighters developed a training routine with an average frequency of 12 times a week, predominantly consisting of aerobic exercises such as running and cycling.

Earlier studies have shown that basal BDNF levels are low in people who do not exercise or participate in sports. BDNF levels seem to have an important relationship with regular

exercise, which has been proven to have positive effects on angiogenesis, neurogenesis, and brain functions (*Nofuji et al., 2012*; *Damirchi et al., 2014*; *Manley et al., 2010*; *Yue et al., 2013*). Moreover, despite being exercise-induced, serum BDNF quantification corresponds to cerebral production, since muscle-produced BDNF does not appear to be released into the bloodstream (*Matthews VB et al., 2009*).

Our results show that BDNF maintained similar levels to baseline after the sparring, which is like one study with Olympic boxers (*Neselius et al., 2013*), but differs from another study which showed elevated serum BDNF concentrations in practicing combat sports at a high level (*Ziemba et al., 2020*). One possible reason the elevation in this protein was not identified in our study would be the time taken to collect the samples, since blood collection from our subjects occurred very early immediately after the sparring session. According to *Manley et al. (2010)* and *Yue et al. (2013)*, it is important to collect biomarker samples at least 24 h after brain injury; however, it cannot yet be stated that early assessments can directly influence the results, since other studies present different assessment moments in their methodology (*Korley et al., 2016b*; *Oztasyonar, 2017*). Therefore, a consensus is necessary to clarify the appropriate moment for this assessment.

Another explanation for the difference is the type of sport practiced. While our participants engage in MMA, in the study by *Ziemba et al. (2020)*, athletes were divided into karate, taekwondo, and a combination of judo, wrestling, and sumo, which involve different forces of attack, defense, and counterattack, directly related to the aggressiveness involved in sports practice and consequently, with the blows given and received to the head. Additionally, the athletes analyzed in this study are high-level sports people, who present a high training frequency compared to the participants in our study.

The nature of our outcomes considering their frequency and especially the intensity of the blows may in turn have not been enough to result in an expression of increased levels of this protein. Finally, our results show a significant reduction in BDNF levels between the moments immediately after the sparring session and 72 h after the sparring session. In clinical practice, BDNF levels are lower in patients who have suffered Traumatic Brain Injure (TBI) when compared to controls on the day of injury, constituting a biochemical parameter which signals a diagnosis of TBI (*Korley et al., 2016a*). Our study evaluated the fighters immediately after the sparring session, and therefore we believe that no reduction in BDNF levels was found at this time due to the time elapsed between the end of the sparring session and the immediate collection.

## UCH-L1

Our results showed significant differences in UCH-L1 levels in athletes before the sparring and immediately after the sparring session, with a decrease of this marker being observed. However, these findings are divergent as most studies show an increase in UCH-L1 levels after concussion (*Brophy et al., 2011*; *Papa et al., 2012*; *Puvenna et al., 2014a*), furthermore, the results coincide with other studies (*Papa et al., 2019*; *Papa et al., 2016*), which showed that a reduction in UCH-L1 can be detected immediately after trauma and remains low at 72 h after the sparring session. According *Mondello et al. (2012)* UCH-L1 levels in athletes

subjected to head impacts can increase up to 6 h after injury, and although they fall rapidly in the following hours, the levels tend to remain significantly elevated for up to 36 h.

About sub-concussion, the study by *Harrell et al. (2022)*, which evaluated rugby players and showed that individuals in the control group did not have a significantly higher serum concentration of UCHL1 than the recreationally active control group. or the pre-game group. Like these findings, the study by *Puvenna et al. (2014b)* in football players showed that serum UCHL1 concentration did not correlate with the number of subconcussive hits to the head. This study also agrees with our findings that UCHL1 levels were unable to distinguish between post-game football players, emergency room patients who suffered mild traumatic brain injury and healthy controls.

The ubiquitin-proteasome complex is associated with cellular vitality by regulating homeostasis. Accumulation of unwanted proteins because of ubiquitin-L1 dependent defective proteolysis may contribute to events underlying the pathogenesis of several major human neurodegenerative diseases such as Alzheimer's and Parkinson's diseases (*Layfield et al., 2001*); therefore, successive brain injuries, accumulated over years in the practice of MMA, can bring about important impairments that affect the physical, cognitive and psychosocial function of individuals (*Riederer et al., 2011*). However, the data from the present study do not support this statement, and longitudinal studies are necessary to reinforce this hypothesis.

## GFAP

Our study found no differences between GFAP levels at the three assessment points of the fighters, although a recent study showed that it was possible to detect GFAP within one hour of injury, peaking after 20 h and declining slowly until 72 h after trauma (*Papa et al., 2019*). We believe that our findings are due to the severity of the brain injury, since GFAP is associated with intense brain trauma (*Okonkwo et al., 2013*) sub-concussive trauma, as it is milder, may not cause typical signs and symptoms, presenting a very low expression as shown in the study by *Papa et al. (2019)*, which possibly explains the lack of significance in our results.

GFAP is a highly regulated protein whose expression is induced by multiple factors such as brain injury and disease (*Eng, Ghirnikar & Lee, 2000*). GFAP is a classical marker for astrocytes known to be induced by brain damage or during degeneration of the central nervous system (*Middeldorp & Hol, 2011*). This protein is a promising biomarker of brain injury in patients with concussion in the acute phase (*Yue et al., 2019*); in addition, GFAP can differentiate individuals with and without abnormalities in computed tomography (CT), being the strongest predictor among the evaluated biomarkers (*Gill et al., 2018*). Cerebrospinal fluid GFAP levels were detected as acute and cumulative effects of head trauma, suggesting minor central nervous system injury (*Neselius et al., 2012*) and plasma GFAP concentrations also correlate with MRI lesion types (*Yue et al., 2019*). However, in our results we did not find significant differences about GFAP in the acute phase of brain injury.

## SCAT5

We found differences between the number of symptoms and the severity of symptoms in the SCAT5 before sparring, immediately after the fight and 72 h after the sparring in the group of fighters. Our findings corroborate the results of Meier et al., who observed significant differences in the SCAT5 in American football players, between the pre-training, six and 24-48 h after the trauma, since during the championship (*Meier et al., 2017a*). However, unlike what was found in the study by *Unnsteinsdottir Kristensen, Kristjánsdóttir & Jónsdóttir (2021)*, in our athletes, sub-concussive symptoms were greater at time 01, that is, before sparring.

We believe that SCAT-5 result is linked, as already mentioned, to the fact that our outcome variable is a subconcussive event and not a concussion. A concussion occurs when a direct or indirect impact is applied to the head with sufficient intensity to generate symptoms (*Belanger & Vanderploeg, 2005*). However, in a subconcussion, there is a transfer of mechanical energy to the brain with sufficient intensity to affect neuronal integrity (*Bailes et al., 2013*), but without resulting in clinical symptoms.

A study of 60 boxers found that most impacts suffered during a 2-minute sparring session were below the threshold for concussion (*Stojsih et al., 2010*). The same could have happened in our study, since the acceleration/deceleration forces, although the intensity for damage to neurons is not yet known (*Montenigro et al., 2015*), exert greater influence on the symptoms presented by the fighter. Additionally, the SCAT-5 is useful immediately after injury in differentiating concussed from non-concussed athletes, but its utility appears to decrease significantly 3–5 days after injury. The symptom checklist, however, does demonstrate clinical utility in tracking recovery.

From a clinical point of view, we found that MMA practitioners had a greater number and severity of symptoms and severity at baseline than sedentary individuals and active, like what was observed by *Petit et al. (2020)*, who found that athletes participating in contact sport reported a severity of symptoms almost twice as high as non-contact athletes. Therefore, these results indicate that even in healthy conditions (free of continuous concussions), practitioners of contact sports tend to have persistent symptoms, which can later influence cognition and neurological changes.

The results of this study reinforce the need for a discussion on primary concussion prevention, aimed at reducing the risk of recurrent injuries and the potential for persistent symptoms. However, further studies addressing different types of sports are needed. According to the Consensus Statement on Concussion in Sport (*Patricios et al., 2023*), healthcare professionals should be encouraged to identify and optimize SRC prevention strategies in their environment. Implementing primary SRC prevention at all levels of sport is a priority that can have a significant impact on public health and reduce the risks of compromising athletes' physical, mental, and cognitive health.

## Strengths and limitations

The small sample size in the study may raise limitations regarding the extrapolation of results. However, the subjects' selection criteria, as well as the calibration of subjective and biochemical measures can minimize potential selection and measurement biases. In

addition, it is worth highlighting the pioneering nature of the study in demonstrating the effects of repeated subconcussive blows to the head in MMA fighters. The use of reliable detection methods and duplicate analysis for all markers reduces the risk of possible errors in biochemical measurements.

This study did not evaluate the intensity of the strikes suffered by the fighters, which may be more related to the severity of brain damage than the number of blows. We believe that some aspects of this study could be improved if this assessment were performed. Moreover, this study provides data on the biomarker levels associated with concussion in MMA fighters, as well as the symptoms referred by them. These data are important for understanding traumatic events in the head, especially in MMA fighting athletes who are submitted to these events weekly.

This was a pioneering study which demonstrated that repeated subconcussive strikes in MMA fighters are associated with reduced levels of UCH-L1 immediately after the fight, as well as reduced levels of BDNF 72 h after the fight. Furthermore, the traumatic events were not able to change the levels of GFAP. These exploratory findings may help in understanding the influence of repeated subconcussive blows on MMA fighters, a population frequently affected by these events.

## CONCLUSION

For clinical practice, our results show that MMA athletes should be submitted to the evaluation of symptoms resulting from subconcussive blows, since craniocervical and vestibular symptoms are subject to physiotherapeutic interventions.

Therefore, it is suggested that future longitudinal studies be carried out. From a preventive point of view, it is worth highlighting the need to assess the influence of the use of head protectors in training as a prevention/protective method, thus minimizing the effects of subconcussive blows on fighters.

## ACKNOWLEDGEMENTS

We thank Dr. Carlos Eduardo Rocha Correia from Federal University of Rio Grande do Norte (Brazil) for his help on the study design and data collection, and Otávio de Souza Marinho Neto and Paulo Leonardo Araújo de Góis Morais from Federal University of Rio Grande do Norte (Brazil) for their assistance on the data collection.

### Funding
This study was financed by the Coordenação de Aperfeiçoamento de Pessoal de Nível Superior - Brasil (CAPES) - Finance Code 001. The funders had no role in study design, data collection and analysis, decision to publish, or preparation of the manuscript.

### Grant Disclosures
The following grant information was disclosed by the authors:

The Coordenação de Aperfeiçoamento de Pessoal de Nível Superior - Brasil (CAPES) - Finance Code 001.

## Competing Interests

The authors declare there are no competing interests.

## Author Contributions

- Nelson Marinho de Lima Filho conceived and designed the experiments, performed the experiments, analyzed the data, prepared figures and/or tables, authored or reviewed drafts of the article, and approved the final draft.
- Sabrina Gabrielle Gomes Fernandes analyzed the data, prepared figures and/or tables, authored or reviewed drafts of the article, and approved the final draft.
- Valeria Costa analyzed the data, authored or reviewed drafts of the article, and approved the final draft.
- Daline Araujo analyzed the data, authored or reviewed drafts of the article, and approved the final draft.
- Clecio Godeiro Jr analyzed the data, authored or reviewed drafts of the article, and approved the final draft.
- Gerlane Guerra conceived and designed the experiments, analyzed the data, authored or reviewed drafts of the article, and approved the final draft.
- Ricardo Oliveira Guerra conceived and designed the experiments, performed the experiments, analyzed the data, authored or reviewed drafts of the article, and approved the final draft.
- Karyna Figueiredo Ribeiro conceived and designed the experiments, performed the experiments, analyzed the data, prepared figures and/or tables, authored or reviewed drafts of the article, and approved the final draft.

## Human Ethics

The following information was supplied relating to ethical approvals (i.e., approving body and any reference numbers):

Research Ethics Committee of the Federal University of Rio Grande do Norte(protocol reference #3.246.228).

## Data Availability

The raw data are available in the Supplemental File.

## Supplemental Information

Supplemental information for this article can be found online at http://dx.doi.org/10.7717/peerj.17752#supplemental-information.

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
