# Peer review of "Levels of biomarkers associated with subconcussive head hits in mixed martial arts fighters"

_PeerJ, doi:10.7717/peerj.17752_

## Round 0.1 · original submission · Major Revisions

The reviewers have made several good suggestions for improving this manuscript. Please pay particular attention to clarity in the methods and results sections.

Reviewer 1 ·

Basic reporting

Thee reporting is generally good but there are some issues with regards to wording/phrasing and sentence structure in places which requires attention. I have highlighted in red on the pdf, as well as adding a comment in some instances, where there are issues with word choice or the spelling/grammar requires some attention.

Experimental design

The general design of the experimental protocol is appropriate for what the authors wanted to evaluate within this study. However, there are some details that are lacking that make the ability to interpret the relevance/significance of the findings. These include;
-for how long prior to the study were the MMA fighters told to abstain from training/exercise?
-how were the sparring sessions conducted? e.g how many rounds, how long were the rounds, what were the size of gloves, were head guards used, was there any indication of how many strikes to the head the participants experienced during the sparring sessions?
-what was the rationale for using bodybuilders as a comparison group?
-in methods, need to state what measures of central tendency and dispersion are presented
-need greater justification for the comparison to the other groups
-need to state what stats tests used for categorical data
-is there additional data on how long avg training sessions were for the MMA fighters? is there also data on how many fights they've had?
-I think some justification as to why measures were taken at 72 hours, when the authors note that many biomarkers return to pre levels at ~36hrs.

Validity of the findings

The opening paragraph of the discussion should give an overview of the main outcomes of the study, with the specifics then addressed in sub-sections. There is some lack of clarity in places due to how information is presented. A major issue in the discussion is the mention of MCP-4 but there is no reporting of this data in the results nor mention of it in the methods? Some greater clarity on whether when you mention "your results being linked..." line 313, are you talking about the SCAT5 data? This could be clarified more clearly.

Annotated reviews are not available for download in order to protect the identity of reviewers who chose to remain anonymous.

·

Basic reporting

Manuscript is professionally presented.
Figures and reporting of statistical test results need to be improved.

Experimental design

Experimental design requires more details, including the nature, timings and durations of the sparring bouts, and what the control groups did instead of a sparring bout.

Validity of the findings

Findings are difficult to evaluate at this stage due to the issues with the reporting of the statistical tests and the results as a whole. More details are needed in the methods section, and potentially a change in the analyses conducted.

Additional comments

To the authors,
Thank you for submitting this work for review, and for undertaking the study in the first place. This is a vital area of research to further our understanding of the effects of combat sports on brain trauma and health, especially in the case of non-diagnosed brain trauma. The use of blood biomarkers may be useful for this, so this paper is well timed. There are, however, some key issues with the manuscript and the data as it is currently presented. Please review my comments below with the good intentions they are written with, as I do hope to see this work in print at some point.
I very much look forward to reading the next submission of this manuscript
Kind regards,
Dr Chris Kirk

INTRODUCTION
Lines 79-81: If using the International Concussion in Sport Conference statement then please use the most up to date statement: https://bjsm.bmj.com/content/57/11/695.abstract . Please also consider including a statement that the majority of head accelerations do not result in a diagnosable concussion, but may still result in brain trauma in the short and long terms (https://doi.org/10.1016/j.nbd.2018.06.016).
Lines 87-88: The number of 228.7/1000 is from Lystad et al and refers to all head injuries including cuts and abrasions. Please rephrase this, or use a different reference to show the rate of head impacts and potential concussions in MMA. The Miarka et al study (reference number 7 in the manuscript may be used for this, or the following: https://doi.org/10.1080/00913847.2020.1856631 which also shows differences in the rate of visually diagnosable concussions between MMA and boxing.
Lines 90-91: Please provide brief examples of what these impairments may present as.
Line 99: Please provide some brief reasons why this might be the case. As this manuscript is focussed on head impacts without a diagnosed concussion, please also provide a brief discussion as to why brain trauma without concussion symptoms are likely to go undiagnosed.
Lines 111-117: This is a very long follow on sentence, with some minor spelling errors. Please check for these errors, and consider rephrasing to make it easier for readers to follow.
Lines 118-121: Please provide evidence that BDNF may be affected by head impacts either with or without diagnosed concussions.
Lines 125-127: repetition of key point from first paragraph.
Lines 128-134: These statements reference the same paper with different notations – please check the manuscript and remove such repeat references. Please also use a more up to date references for the number of head impacts in MMA bouts. Buse et al’s. paper is from almost 20 years ago, and there have been widespread changes in the profile of MMA performance even in the last decade. Suggest: DOI: 10.1519/JSC.0000000000001804
Lines 135-138: Suggest including brief discussion of recent pilot data that provides some of the first data potentially measuring the effect of repeated ‘sub-concussive’ head impacts in MMA sparring to provide a field based diagnostic tool: https://www.mdpi.com/2411-5150/7/2/39

METHODS
Sparring bouts: How many rounds were completed, how long did these rounds last, and how long was the recovery time between each bout? What equipment did the participants wear for these bouts? How long before and after the bouts did each measurement take place? Did any of the participants experience a diagnosable concussion in these bouts?
Control groups: What did the two control groups do instead of a sparring bout? What were the timings of their measurements and were these matched to the MMA participant’s timings?
Lines 157-159: Please make this section clearer – who was this information provided to and how was it provided?
Figure 2: Please state in the figure that ‘symptom evaluation’ was application of the SCAT-5. If possible, please refer to the specific section of the SCAT-5 used by name, to avoid readers thinking that the whole SCAT-5 is being reported.
Ethics: Suggest moving this to the start of the methods section, and if possible provide the date ethical approval was granted.
Lines 187-188: Can it be guaranteed that the participants had not experienced any head impacts in the 5 days prior to the first blood draw?
Lines 192-194: Please make this section clearer and more specific – for example what does the phrase “about 450 nm” refer to?
Lines 202-203: Which variables are being classified as ‘categorical variables’ here? Why was a χ2 test used for these?
Lines 203-209: Why are separate ANOVAs used at each time point and for each group? Why not use a three-way repeated measures ANOVA between all time points and groups to avoid error inflation due to multiple testing? Please either provide a suitable justification for multiple testing, or alter the analyses to a repeated measures ANOVA between the three groups and three time points together.
Statistics: Please also calculate and report effect sizes for each test to provide a more detailed evaluation of the differences between groups and for inclusion in future meta analyses. Suggest either η 2 or ω2.

RESULTS
Please provide full F values and df for each test to allow full evaluation of the results by the readers.
Figures: The figures are currently not set out in a way that allows easy comparison between groups and time points. Please redraw these figures to include all three groups and timepoints on one plot for each of the variables. So one figure showing the BDNF data for all three groups at all three time points, one figure for GFAP, one figure for UCHL-1, etc. Suggest using line plots for these instead of bar charts.
Lines 221-225: Please provide the SCAT-5 results for all time points measured.
Lines 229-232: Were these differences between groups for each time point? Or were these differences only for the sparring group?

DISCUSSION
Lines 250-251: Why might this be the case? Please provide references to support BDNF values to be dependent on activity levels.
Lines 260-271: Why might this be the case? What sampling time is associated with BDNF being indicative of brain trauma? Please provide explanations and evidence for these statements.
Lines 275-276: But is there any evidence of ECH-L1 changing in relation to ‘sub-concussive’ head trauma?
Lines 279-286: Do the reported data provide any evidence for these effects? Do the reported data match previous data or not?
Lines 292-296: Why would this have affected the results? Please explain.
Lines 297-304: Similar to the comments above, how does this discussion relate to the reported data? Do the reported data match previous data or not?
Lines 313-321: This is a key discussion point about the utility of the SCAT-5 in that these data demonstrate that it is only useful for occasions where a concussion is likely, i.e., it has been witnessed and there are diagnosable symptoms. As such, it is probably not a useful tool for measuring the effects of ‘sub-concussive’ head trauma such as that seen in MMA sparring.
Lines 342-344: MCP-4 has not been measured in this study, please correct this statement. Please also revisit the results section and figures to make it clearer whether the MMA group were statistically different to the control groups or not. If the MMA group’s data increased but was not different to the control groups then we cannot say that these changes were due to the sparring.
Lines 350-351: Please change this statement – these is no data provided here that suggests these markers can identify ‘lesions’.

---

## Round 0.2 · Minor Revisions

Reviewer 1 has asked that you carefully proof read the manuscript to improve clarity. Reviewers 2 has made several suggestions that will help to improve the manuscript.

Reviewer 1 ·

Basic reporting

Thank you for the changes. There are still some minor issues with wording in places that could do with addressing.
Line

Experimental design

Thank you for clarifying and addressing the issues raised.

Validity of the findings

No comment

Additional comments

There are just some minor issues with wording/phrasing in places that need checking.

·

Basic reporting

Reporting is more clear on this submission than on the previous, though the figures cane be improved (details provided in comments of review).

Experimental design

Design is appropriate, though the wording of parts of the methods section could make it more clear.

Validity of the findings

Findings appear valid, though effect sizes need to be reported.

Additional comments

To the authors,
Thank you for responding to the first review and re-submitting your work. The data are now much clearer and easier to evaluate in the context of the experiment conducted. The additions to the methods section also make the study easier to follow and understand. There are still some relatively minor changes and amendments to make in the methods and results sections, and I have also made some suggestions about the Discussion section that might strengthen the arguments being made.
Thanks,
Dr Chris Kirk

Please check throughout for grammar and typing errors potentially caused by editing after the first review.

INTRODUCTION
Lines 118-129: Suggest discussing the cognitive and functional deficits reported by these studies. This would help support the need for developing better measurements of brain trauma in the absence of concussion. These studies (refs 35-39) all show that MMA athletes experience multiple head impacts during training and competition, very few of which result in a diagnosable concussion. The cognitive deficits the athletes experience, however, show that brain trauma has still occurred and needs to be measured. This would support the need for this study and other like it.

METHODS
Lines 170-171: It’s still unclear as to what the control groups were doing and when their measurements were taken. Did the control groups only provide data once? If so, when was this? Did the control groups perform any activities at all during the study or were they asked to remain inactive?
Lines 195-197: Please change the wording of this to state that the participants were requested to avoid any head impacts or head trauma for 5 days prior, as you cannot be certain that they did avoid this.
Lines 208-209: Please make it clear that the χ2 test is only being used to compare the descriptives between groups.
Lines 212-213: This suggests that only the MMA group’s biomarkers were collected and analysed, but they are being compared to the control groups. Please reword.
Lines 213-214: Please reword this section to make it clear that all three groups were being compared here – it currently reads as though an ANOVA was used for the MMA group and a separate ANOVA was used for the other two groups. In addition, given that three groups are being compared shouldn’t this be a three factor/three way ANOVA rather than a one way ANOVA? Please also see comment in the Methods for Lines 170-171 about when the control group’s data were collected – at what time point are the groups being compared? Please calculate and report effect sizes for all ANOVAS to provide a full contextualisation of the data and to allow for inclusion in future meta-analyses. Suggest either ω2 or ηp2.
Please provide details of any power analyses completed prior to the study commencing.

RESULTS
Please report the full statistical results for all tests, including the F and df values. Please also make it clear which of the results are for the overall ANOVA, and which are the post hoc results.
Figures: Please provide values on the y axis of each figure to indicate the scale of pg∙ml being reported for the sample. Please also provide error bars on each column – and I suggest changing these figures to box plots with the group labels on the x axis rather than bar plots as these measurements are point estimates in time rather than overall quantities.
Table 1: Please provide units for ‘Physical Activity Frequency’ – is this hours per week, sessions per week, etc.?

DISCUSSION
Lines 262-264: Please provide this information in the results or participant characteristics table – their exercise frequency is reported, but not the general content or mode of their training. As this could have an influence on BDNF levels (as discussed by the authors of this paper) this information should be reported.
Lines 271-276 and lines 287-289: Could another explanation be that the participants in Ref 55 had diagnosed concussions, whereas the participants in Ref 54 and in the present study did not? Might it be that BDNF is related to concussion but not head impacts without concussion?
Lines 303-305 and lines 312-313: But it does show changes following sparring – might it be due to the relatively small sample size rather than no effect? This is where inclusion of the error bars on the plots may become important.
Lines 352-357: In my opinion this is the most important finding of the current study. Suggest some discussion as to how this might be used in practice and in further research to monitor potential changes in athlete cognitive health.

---

## Round 0.3 · accepted · Accept

I have assessed the latest version of the manuscript and, in my opinion the authors have addressed all of the reviewers' comments. All comments were minor and I believe the manuscript is now ready for publication.